

Water-soluble iron correlation to primary speciated organics in low-emitting vehicle exhaust
Joseph R. Salazar*, Benton T. Cartledge*, John P. Haynes*, Rachel York-Marini*, Allen L
Robinson[‡], Greg T. Drozd[€], Allen H. Goldstein[¥], Sirine C. Fakra[¢], Brian J. Majestic*
*University of Denver, Department of Chemistry and Biochemistry
[‡]Carnegie Mellon University, College of Engineering
[¥]University of California, Berkeley Department of Civil and Environmental Engineering
[€]Colby College Department of Chemistry
[¢]Advanced Light Source, Lawrence Berkeley National Laboratory, Berkeley, CA 94720
*Correspondence to:* Brian J. Majestic (brian.majestic@du.edu)
**Abstract**
Iron is the most abundant transition element in airborne PM, primarily existing as Fe(II)
or Fe(III). Generally, the fraction of water-soluble iron is greater in urban areas compared to
areas dominated by crustal emissions. To better understand the origin of water-soluble iron in
urban areas, tail-pipe emission samples were collected from 32 vehicles with emission
certifications of Tier 0, low emission vehicles (LEV I), tier two low emission vehicles (LEV II),
ultralow emission vehicles (ULEV), superultra-low emission vehicles (SULEV), and partial-zero
emission vehicles (PZEV). Components quantified included gases, inorganic ions, EC/OC, total
metals and water-soluble metals. In addition, naphthalene and various classes of C12-C18
intermediate volatility organic compounds (IVOC) were quantified for a subset of vehicles:
aliphatic, single ring aromatic (SRA), and polar (material not classified as either aliphatic or
SRA). Iron solubility in the tested vehicles ranged from 0 – 82% (average = 30%). X-ray
absorption near edge structure (XANES) spectroscopy showed that Fe(III) was the primary
oxidation state in 14 of the 16 tested vehicles, confirming that the presence of Fe(II) was not the
main driver of water-soluble Fe. Correlation of water-soluble iron to sulfate was insignificant, as
was correlation to every chemical component, except to naphthalene and some C12- C18 IVOCs



with $R^2$ values as high as 0.56. A controlled benchtop study confirmed that naphthalene, alone,
increases iron solubility from soils by a factor of 5.5 and that oxidized naphthalene species are
created in the extract solution. These results suggest that the large driver in water-soluble iron
from primary vehicle tail-pipe emissions is related to the organic composition of the PM,
indicating the organic fraction of the PM influences the behavior and solubility of iron.
**1. Introduction**
Iron has been identified as a limiting nutrient for phytoplankton in approximately half of
the world's oceans, with deposition from the atmosphere as the major source (Moore and Abbott,
2002; Sholkovitz et al., 2012). Phytoplankton is one of the controlling factors of fixed nitrogen in
many parts of the oceans and, consequently, plays a major role in the ocean's biogeochemical
cycles (Baker et al., 2006; Chen and Siefert, 2004; Kraemer, 2004; Shi et al., 2012; Tagliabue et
al., 2017). Also, water-soluble iron fractions are linked to the creation of reactive oxygen species
(ROS) in lung fluid and in environmental matrices through Fenton chemistry (Hamad et al.,
2016). These ROS impart oxidative stress on the respiratory system, contributing to various
health effects (Landreman et al., 2008; Park et al., 2006; Verma et al., 2014).
Annually, approximately 55 Tg of iron enters the atmosphere from crustal sources (Luo
et al., 2008). Of this, 14-16 Tg are deposited into the ocean, impacting the marine life and
influencing the ecosystems (Gao, 2003; Jickells et al., 2005). Typically, airborne iron from
crustal sources ranges from 0.05-2% water-soluble of the total iron (Bonnet, 2004; Sholkovitz et
al., 2012). Relative water-soluble iron in urban environments is higher, ranging from 2-50% of
the total (Majestic et al., 2007; Sedwick et al., 2007; Sholkovitz et al., 2012). It is suggested that
combustion sources including fossil fuel burning, incinerator use and biomass burning may be a
large contributor to the water-soluble iron fraction, contributing 0.66-1.07 Tg a$^{-1}$ of water-soluble

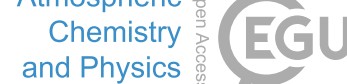

iron and this iron has been correlated to anthropogenic sources (Chuang et al., 2005; Luo et al.,
2008; Sholkovitz et al., 2009). Even though total iron emissions from combustion sources are
small in comparison to crustal sources, the relative insolubility of crustal iron leads to the
possibility that combustion sources contribute 20%-100% of water-soluble iron into the
atmosphere (Luo et al., 2008; Sholkovitz et al., 2012).

Previous studies in tunnels and parking structures have reported iron ranging from five to

approximately 3,500 ng m$^{-3}$, revealing that brake wear, tire wear, resuspended road dust, and tail
pipe emissions can be important sources of trace elements (Kuang et al., 2017; Lawrence et al.,
2013; Li and Xiang, 2013; Lough et al., 2005; Park et al., 2006; Verma et al., 2014). Within the
engine, iron is likely to result from combusted gasoline, which has pre-combusted concentrations
ranging from 13-1000 µg L$^{-1}$ (Lee and Von Lehmden, 1973; Santos et al., 2011; Teixeira et al.,
2007). Additionally, computational models of combustion in engines suggest that iron emissions
could originate from the fuel injector nozzle inside the engine block (Liati et al., 2015).

There are many different factors that may contribute to water-soluble iron and, as a result,

several different hypotheses have been developed relating to how iron is solubilized in ambient
atmospheres. First, correlation of ambient iron to sulfates in ambient aerosols suggest the
possibility of iron solubilization (Desboeufs et al., 1999; Hand et al., 2004; Mackie et al., 2005;
Oakes et al., 2012b). However, laboratory studies investigating the heterogeneous chemistry of
iron have not shown any change in iron water-solubility, speciation, or oxidation state upon
exposure to gaseous $SO_2$ (Cartledge et al., 2015; Luo et al., 2005; Majestic et al., 2007; Oakes et
al., 2012a). A second hypothesis is that particle-bound iron oxidation state may control iron
water solubility.  Thus far, the limited field studies have been unable to show that iron oxidation
state is correlated to iron's resulting water solubility, as the majority of iron found in aerosol



particles is in the less soluble Fe(III) oxidation state (Luo et al., 2005; Majestic et al., 2007;
Oakes et al., 2012a). A third, broad, iron solubilization hypothesis emphasizes an iron-organic
interaction (Baba et al., 2015; Vile et al., 1987). For example, a significant increase in water-
soluble iron is observed in the presence of oxalate and formate in ambient aerosols and in cloud
droplets (Paris et al., 2011; Zhu et al., 1993). Other studies have suggested that the photolysis of
polycyclic aromatic hydrocarbons leads to reduced iron, which may result in greater iron water
solubility (Faiola et al., 2011; Haynes et al., 2019; Pehkonen et al., 1993; Zhu et al., 1993).
In this study, we explore all three hypotheses (bulk ions, iron oxidation state, and organic
speciation) in relation to iron solubility. Specifically, we examine the water-soluble iron emitted
from 32 light duty gasoline vehicles with certifications of Tier 0, low emission vehicle (LEV I),
tier two low emission vehicles (LEV II), ultralow emission vehicles (ULEV), superultra-low
emission vehicles (SULEV), and partial-zero emission vehicles (PZEV). The total and water-
soluble trace elements are compared to the ions, gaseous compounds, and organic emissions
from the same vehicle set. Additionally, we acquired data on the emitted iron oxidation states on
the exhaust particles. From this data set, real tail-pipe emission samples were explored to
discover how various components of automobile exhaust affect the water solubility of iron.
**2. Materials and Methods**
**2.1. Sample Collection**
Exhaust samples from 32 gasoline vehicles were collected at the California Air Resources
Board (CARB) Haagen-Smit laboratory over a six-week period. Standard emission test results
from this campaign have been reported previously (Saliba et al., 2017). A description of the
dynamometer, emission dilution system, and instrumentation used in the vehicle set up is
provided elsewhere (May et al., 2014; Saliba et al., 2017). Briefly, each vehicle was tested on a
dynamometer using the cold-start Unified California (UC) Drive Cycle or the hot start Modal
Arterial Cycle 4. Emission samples were collected using a constant volume sampler from which
a slipstream of dilute exhaust was drawn at a flow rate of 47 L min$^{-1}$. The particulate exhaust
emissions were then collected on the pre-cleaned Teflon filters. The Teflon filters were stored in
a freezer until extraction and analysis was performed. Filter holders were maintained at 47°C
during sampling as per the CFR86 protocol.

The vehicles were recruited from private citizens, rental car agencies, or part of the Air

Resource Board fleet. The vehicles tested were categorized by model years (1990-2014), vehicle
type (passenger car and light-duty trucks), engine technologies (GDI and PFI), emission
certification standers (Tier1 to SULEV), make, and model. All vehicles were tested using the
same commercial gasoline fuel which had a 10 % ethanol blend and a carbon fraction of 0.82
(Saliba et al., 2017).

Gases (CO, $CO_2$, $CH_4$, NO, and $NO_2$) and total hydrocarbons (THC) were collected into

heated Tedlar bags by UC Drive Cycles. Analysis of CO and $CO_2$ was measured by
nondispersive infrared detectors (IRD-4000), $CH_4$ by gas chromatography-FID, $NO_x$ by
chemiluminescence (CLD 4000) and THC by FID (Drozd et al., 2016; Saliba et al., 2017).
Particle phase emissions were collected using three sampling trains operated in parallel off of the
end of the CVS dilution tunnel. Train 1 contained a Teflon filter (47 mm, Pall-Gelman, Teflo
R2PJ047). Train 2 contained two quartz filters (47 mm, Pall-Gelman, Tissuquartz 2500 QAOUP)
in series. Train 3 contained an acid-cleaned Teflon filter followed by a quartz filter (47 mm,
Teflo, Pall Life Sciences, Ann Arbor, MI). The Teflon filter in Train 1 was analyzed by ion
chromatography for water-soluble anions and cations and these data are presented elsewhere
(Hickox et al., 2000). Train 2 included two parallel sets of Tenax-TA sorbent tubes (Gerstel)





downstream of the Teflon filter. The first set was 2 tubes connected in parallel. One of these
tubes was used to collect emissions during the cold start phase of UC (the first five minutes,
commonly referred to as bag 1).  The other tube was used to sample emissions during the
combined hot-running and hot start phases of the UC (bags 1 and 2). The second set of sorbent
tubes was connected in series to collect emissions over the entire UC test. The flow rate was 0.5
L min$^{-1}$ through each Tenax tube. The Teflo filter in Train 3 was used for total and water-soluble
trace element analysis and particle-bound iron oxidation state and is the focus of this study.
**2.2. Materials Preparation**

All vessel cleaning and analytical preparation for the trace elements was performed under

a laminar flow hood with incoming air passing through a high efficiency particulate air (HEPA)
filter. All water used was purified to 18.2 MΩ-cm (Milli-Q Thermo-Fisher Nanopore).  Fifteen
and 50 mL plastic centrifuge vials, Petri dishes (Fisher), Teflon forceps (Fisher), syringe
(Fisher), nitro cellulose paper (Fisher), and syringe cases (Life Sciences Products) were prepped
by an acid cleaning process. For the plastic centrifuge vials, Petri dishes, Teflon forceps, syringe,
and syringe cases this involved 24-hour soaks in a 10% reagent grade nitric acid bath followed
by 10% reagent grade hydrochloric bath then a 3% trace metal grade nitric acid (Fisher) resting
bath with MQ rinses before, after and between each step. The nitro cellulose paper was cleaned
by soaking in 2% HCl for 24 hours then rinsing with MQ water. Then, 2% HCl and MQ water
were pushed through the filter. Teflon beaker liners were cleaned by an acetone rinse, then an
overnight bath of 100% HPLC-grade acetonitrile and a final overnight bath of 5% trace-metal
grade nitric acid. 0.20 micron syringe filters (Whatman, Marlborough, MA) were prepared with
10% trace-metal grade hydrochloric acid, MQ water and 5% nitric acid rinse. All materials were

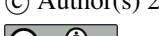



handled with powder free nitrile gloves (Fisher), double bagged, and handled inside a
polypropylene laminar flow hood (NuAire, Plymouth, MN).

The 47 mm Teflon filters were cleaned by submerging them in 10% trace metal grade

nitric acid and rinsing with MQ water. The filters were then stored in the acid cleaned Petri
dishes and sealed with Teflon tape for storage.
**2.3. Water-soluble metals sample preparations**

Water-soluble elements were extracted for 2 hours from the Teflon filter on a shaker table

in 10 mL of MQ water. The water extract was filtered with 2 µm pore size nitro cellulose filters.
The Teflon filter and the nitro cellulose filters were saved for total metals digestion. The water-
soluble element extract was acidified to 5% trace-metal grade nitric acid and 2.5% trace-metal
grade hydrochloric acid to be analyzed by inductively coupled plasma mass spectrometry (ICP-
MS, Agilent 7700).
**2.4. Sample preparation for total elemental analysis**

After the polymethylpentene ring was removed from the Teflon filters and ~3%

(measured exactly) of the filters cut and saved for X-ray absorption near edge structure
(XANES) spectroscopy, the Teflon and the nitro cellulose filters for each sample were placed
together into a microwave digestion vessel. To each digestion vessel, 750 µL of concentrated
trace metal grade nitric acid, 250 µL of concentrated trace grade hydrochloric acid, 100 µL of
concentrated trace grade hydrofluoric acid, and 100 µL of 30% hydrogen peroxide was added.
These samples were digested (Ethos EZ, Milestone Inc) according to the following a temperature
program: 15-minute ramp to 200 °C, then held at 200 °C for 15 minutes, and a 60-minute cooling



period.(Cartledge and Majestic, 2015) The samples were cooled to room temperature for 1 hour
and the solution was diluted to 15 mL with MQ water and analyzed via ICP-MS.
**2.5. Elemental analysis**

Blank filters and standard reference materials (SRMs) were digested alongside the

exhaust samples using the same digestion process described above. Three SRMs were used to
address the recoveries of our digestion process: urban particulate matter (1648a, NIST), San
Joaquin Soil (2709a, NIST), and Recycled Auto Catalyst (2556, NIST). The recoveries of the
SRMs were between 80-120%. The elements analyzed included Na, Mg, Al, K, Ca, Ti, V, Cr,
Mn, Fe, Co, Ni, Cu, Zn, As, Se, Rb, Sr, Mo, Rh, Pd, Ag, Cd, Sb, Cs, Ba, Ce, Pt, Pb, U. Indium
(~1 ppb) was used as an internal standard and a He collision cell was used to remove isobaric
interferences.
**2.6. XANES Spectroscopy**

X-ray absorption near-edge structure (XANES) and X-ray fluorescence (XRF) data for

16 vehicle exhaust samples were collected at the Advanced Light Source Microprobe beamline
(10.3.2), Lawrence Berkeley National Laboratory, Berkeley, CA (Marcus et al., 2004). To locate
iron spots on the filters, a broad XRF elemental map of each sample was acquired at 10 keV
using 12 µm by 12 µm pixel size and 50 ms dwell time per pixel. µXRF spectra were
simultaneously recorded on each pixel of the map. Iron oxidation state and iron-bearing phases
were investigated using iron K-edge extended XANES. The spectra were recorded in
fluorescence mode by continuously scanning the Si (111) monochromator (Quick XAS mode)
from 7011 to 7415 eV. The data were calibrated using an iron foil with first derivative set at
7110.75 eV (Kraft et al., 1996). All data were recorded using a seven-element solid state Ge



detector (Canberra, ON). The spectra were deadtime corrected, deglitched, calibrated, pre-edge
background subtracted and post-edge normalized using a suite of LabVIEW custom programs
available at the beamline (Marcus et al., 2008). To rapidly survey iron oxidation state, a valence
scatter plot was generated from normalized XANES data using a custom Matlab code and a large
database of iron standards (10.3.2 XAS database) (Marcus et al., 2008). Least-square linear
combination fitting (LCF) was subsequently performed in the range 7090 to 7365 eV to confirm
iron valence and further identify the major mineral groups present. The best fit was chosen based
on 1) minimum normalized sum-square value (NSS=$100 \times [\sum(\mu_{exp} - \mu_{fit})^2 / \sum(\mu_{exp})^2]$), where the
addition of a spectral component to the fit required a 10% or greater improvement in the NSS
value,  and 2) on the elements detected in the µXRF spectrum recorded on each XANES spot.
The uncertainty on the percentages of species present is estimated to be ±10%.
**2.7. Organic Speciation**

A subset (10) of the 32 samples were quantified for IVOC using electron impact ionization

with methods similar to that of Zhao et al., except adapted for GCxGC methods (Zhao et al.,
2015, 2016). IVOC material was classified into three categories: aliphatic, single ring aromatic
(SRA), and polar (Drozd et al., 2019). Classification within these three classes of compounds
was determined by differences in second dimension retention time (polarity space) and by mass
spectral characteristics in our GCxGC-MS analysis. All three classes of compounds were
quantified by either compound specific calibration using known standards or relating total ion
chromatogram (TIC) signals to calibration standards of similar volatility and polarity. In
GCxGC, the TIC signal corresponds to a blob, or a region in volatility and polarity retention
space. The GC-Image software package was used to create blobs from 2D chromatograms.
Compounds were quantified by relating their TIC signal to that of the nearest standard in terms





of polarity and volatility. Volatility bins were defined that are evenly spaced with their center
elution times corresponding to each *n*-alkane. TIC blobs were quantified using the calibration for
the available standard of similar polarity in the same volatility bin.
**2.8. Emission Factor Calculations**

Emissions data are presented as fuel-based emission factors (EF). Emission factors are

calculated as the amount of analyte emitted by mass per gram of fuel emitted.
$$\text{EF}_i(g\ g-fuel^{-1}) = \Delta m_i \frac{x_c(g)}{\Delta CO_2(g) + \Delta CO(g) + \Delta THC(g)}$$

$\Delta CO_2$, $\Delta CO$, and $\Delta THC$ are the background corrected carbon concentration of $CO_2$, CO, and
THC (Drozd et al., 2016; Goldstein et al., 2017), respectively. $x_c$ is the fuel carbon mass
fraction of 0.82. $\Delta m_i$ is the blank subtracted concentrations of species *i*.
**2.9. Naphthalene and Iron Benchtop Study**

To better understand the production of soluble iron during the water extraction process, a

bench-top study was performed using three varying forms of iron with naphthalene. The iron
stock solutions/suspensions included: 1) standardized San Joaquin soil (NIST SRM 2709a)
containing 25 ppm total iron (soluble + insoluble) iron to determine the effects of crustal iron, 2)
iron(II) sulfate to a concentration of 25 ppm to examine the effect of a soluble iron(II) source,
and 3) iron(III) sulfate to examine a source of soluble iron(III). In parallel, 100 mg of
naphthalene crystals were added to 200 mL of MQ water. For the experiment, 100 mL of the
naphthalene suspension and 1 mL of the iron suspension were added to Teflon liners (250 ppb
iron total), which were inserted into a jacked glass beaker temperature controlled to 25 °C. After
16 hr of stirring, 2 ml were filtered (0.2 μm) and acidified to 5% nitric acid. Soluble iron released

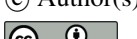



from the soil both in the presence and absence of naphthalene was analyzed by ICP-MS.
Chemical changes in naphthalene in the presence and absence of iron were monitored by HPLC.
**3. Results and Discussion**
*3.1. Total and water-soluble element exhaust concentrations*
Table 1:

Emissions of ions, organic species, gaseous species, and EC/OC from these tests have

been published previously (Drozd et al., 2016, 2019; Goldstein et al., 2017; Saliba et al., 2017).
In order to obtain a better understanding of the factors that influence iron solubility, we compare
these with the total elements, trace elements, and iron oxidation state measurements. Generally,
the elements with the highest EF are the lighter crustal elements Ca, Al, and Fe, with average EF
200, 100, and 80 µg kg-fuel$^{-1}$ (Table 1), respectively. Iron has the third highest average EF of all
the elements and the highest of all transition elements, ranging from 0 – 200 µg Fe kg-fuel$^{-1}$.
This is followed by three first row transition elements: Zn, Cu, and Ni with the respective
average EF of 60, 20, and 5 µg kg-fuel$^{-1}$. Other notable elements include Rh, Pd and Pt, likely
originating from the catalytic convertor, with the respective average EF of 0.05, 0.7, and 0.04 µg
kg-fuel$^{-1}$. Toxic elements include Cr, Pb, Mo and Sb with respective EF 5, 0.8, 5 and 0.2 µg kg-
fuel$^{-1}$. A previous study has shown that various elements are enriched in used motor oil such as
copper, zinc, manganese, iron and lead which could originate from engine wear (Majestic et al.,

2009).

Table 1 also shows the EF for the water-soluble fraction of the trace elements. The water-

soluble EF for iron ranges from 0-150 µg kg-fuel$^{-1}$; or 0-82% of the total. At 20 µg kg-fuel$^{-1}$,
average water-soluble iron was the third largest EF of all elements. There were relatively high



emissions of a few other water-soluble elements such as Ca with an average EF of 200 µg kg-
fuel$^{-1}$ and Zn with tailpipe emissions averaging 40 µg kg-fuel$^{-1}$.
Table 2:

Only a few studies report tailpipe emissions (i.e., dynamometer testing) of trace elements

for diesel and gasoline-powered passenger cars and even fewer which have reported iron water
solubility in low emitting vehicles (Tier1 to SULEV) (Norbeck et al., 1998; Schauer et al., 2002).
Table 2 compares the average exhaust PM composition and trace elements km$^{-1}$ in this study to
literature values for other passenger vehicles, including one diesel and three gasoline exhaust
studies.  For all elements, the per km emissions were greater in the diesel cohort, relative to the
gasoline vehicles.  Compared to previous studies, the trace elements emitted from older gasoline
passenger vehicles resulted in an order of magnitude higher emissions for all elements, except
for aluminum, which only showed a factor of ~2 increase in older vehicles (Table 2). Iron shows
a large range in the three studies of gasoline vehicles, ranging from 8.3-280 µg km$^{-1}$, compared
to the 0-62 µg km$^{-1}$ measured in this study.

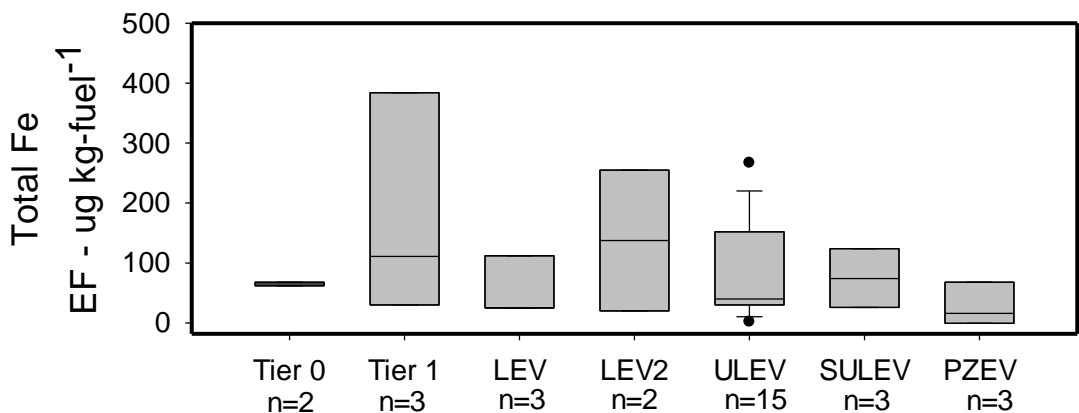





Figure 1: Total iron from the 32 vehicles tested reported in EF ($\mu$g kg-fuel$^{-1}$). The center black
line represents the median value and the edges of the boxes represent the 25$^{th}$ and 75$^{th}$ percentiles
while the whiskers extent are the 10$^{th}$ and 90$^{th}$ percentiles.

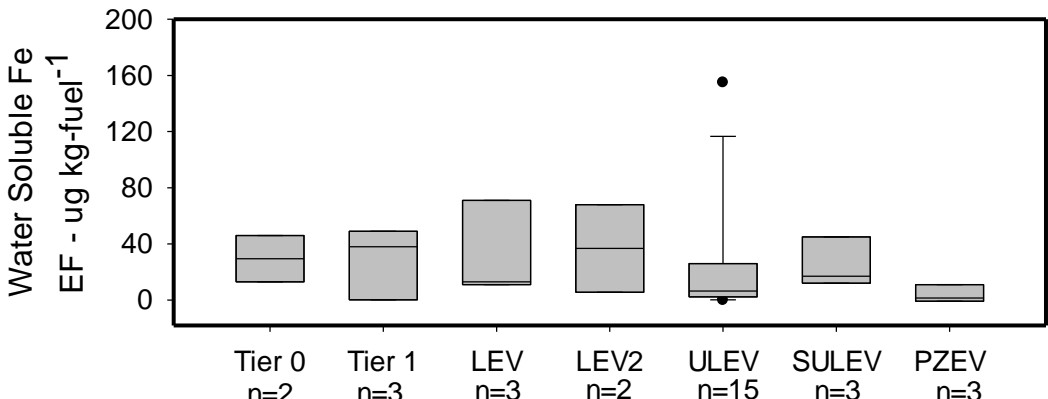


Figure 2: Water-soluble iron from the 32 vehicles tested reported in EF ($\mu$g kg-fuel$^{-1}$). The center
black line represents the median value and the edges of the boxes represent the 25$^{th}$ and 75$^{th}$
percentiles while the whiskers are the 10$^{th}$ and 90$^{th}$ percentiles.

The large ranges in iron solubility of the previous studies led us to explore and compare

the newer emission certification standard (Figure 1 and 2). Total iron did not trend strongly with
emission certification standard, although, on average, total iron is less in the Tier 0 and LEV
vehicles. Water-soluble iron shows a small average decrease of approximately 5 $\mu$g kg-fuel$^{-1}$
between ULEV and SULEV vehicles, and a further average decrease for the PZEV vehicles of
3.9 $\mu$g kg-fuel$^{-1}$.
*3.2. Iron correlations with bulk exhaust components and iron oxidation state*

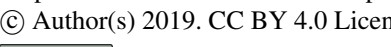



To explore what factors and if any exhaust components are associated with the presence
of water-soluble iron, linear regression analyses were used to compare soluble iron to different
chemical species in the exhaust. Solubility from the direct exhaust was explored by comparing
the EFs of both sulfate and nitrate to iron, and water-soluble iron was not correlated to either of
these species (SI1). The EFs for water-soluble iron and $CO_2$ showed no correlation, suggesting
that overall fuel use was not an important factor for water-soluble iron production (SI1). Total
iron was correlated to the water-soluble iron indicating the total amount of iron may have an
impact on soluble iron (SI2). Finally, to evaluate if water-soluble iron and overall particulate
carbon relate, the EFs for elemental carbon (EC) and organic carbon (OC) were compared to that
of soluble iron and, again, no correlation was observed (SI1).

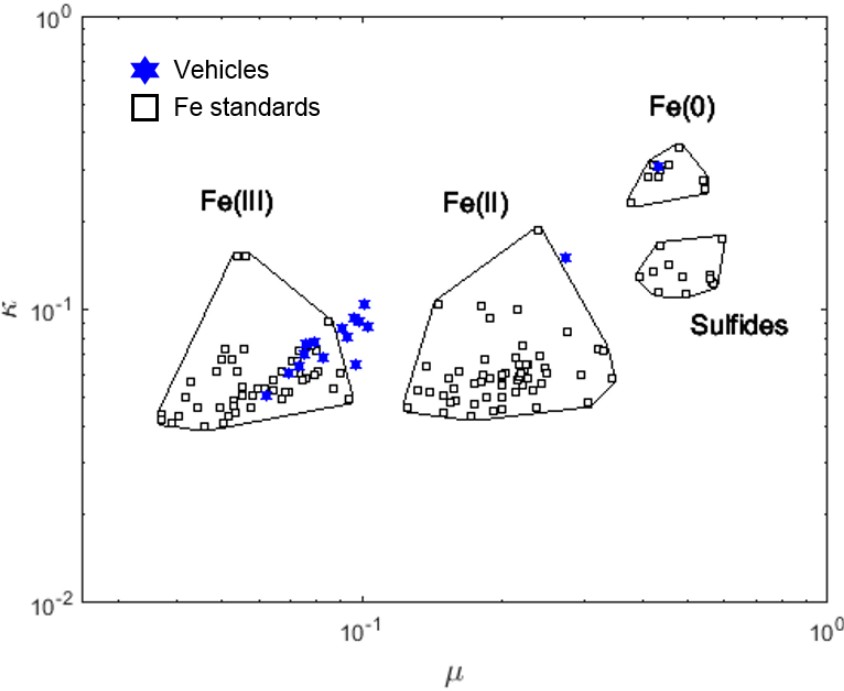






Figure 3: Fe valence scatter plot generated from Fe K-edge XANES data where κ and µ are
normalized absorbance values at 7113 eV and 7117.5 eV respectively. Empty black squares
represent Fe standards of known valence while blue-filled stars represent vehicle exhaust
samples.

As no correlation between water-soluble iron and bulk chemical species was observed

(SI1 and SI3), the importance of the particle-bound iron oxidation state was investigated. Since
Fe(II) is known to be more soluble than Fe(III), the expectation was that exhaust samples having
a large Fe(II) character would have a greater iron solubility, relative to those containing Fe(III)
or to Fe(0) (Stumm and Morgan, 1996).  Figure 3 presents a scatter plot of the iron valence in 16
of the exhaust samples, compared with iron-bearing standards of known valence. This valence
plot is generated from iron K-edge XANES data where parameters κ and µ are defined as
normalized absorbance values at 7113 eV and 7117.5 eV, respectively. We observe that the
exhaust-iron is primarily in the Fe(III) oxidation state, except for two vehicles: sample 11,
dominated by Fe(0) and sample 15, containing a combination of Fe(0) and Fe(III) (SI4). Sample
11 is an extreme case, having 0 % iron solubility and highly elevated amount of EC at 305 µg
kg-fuel$^{-1}$ (study average = 78 µg kg-Fuel$^{-1}$). The presence of Fe(0) is consistent with high EC, as
both observations suggest a lack of oxidation during the combustion and emission process.
While the valence plot (Figure 3) put sample 15 as mostly Fe(II), the LCF actually showed that it
was a mixture of Fe(0) and Fe(III). And, this sample contained only 10% water-soluble iron, less
than the cohort average. The study-wide solid phase iron oxidation state is primarily Fe(III) or
mixed oxidation state (Fe(III) and Fe(0)) (see Figure 3), averaging about 30% water-soluble iron,
well above the crustal background.





LCF XANES fitting (SI4) showed Fe(III) oxides and oxyhydroxides as the dominant group,
followed by Fe(III) sulfates and iron silicates (SI4). Hematite ($\alpha$-Fe$_2$O$_3$) and maghemite ($\gamma$-
Fe$_2$O$_3$) were the most consistently detected Fe(III) oxides. Iron was detected in all samples, with
Zn, Cr and Cu the main other elements detected in nearly all samples (detection of low-Z
elements below sulfur or high-Z elements above zinc was not possible in our experimental
conditions). Overall, these results strongly suggest that the main driver of water-soluble iron is
not associated with the particle-bound iron oxidation state. Further investigation for the LCF
XANES fitting showed that 34% of iron speciated was Fe(III)-oxyhydroxides associated with
organic material, leading to the investigation of longer chain IVOC and naphthalene (SI6).
*3.3. Iron solubility and speciated organics*

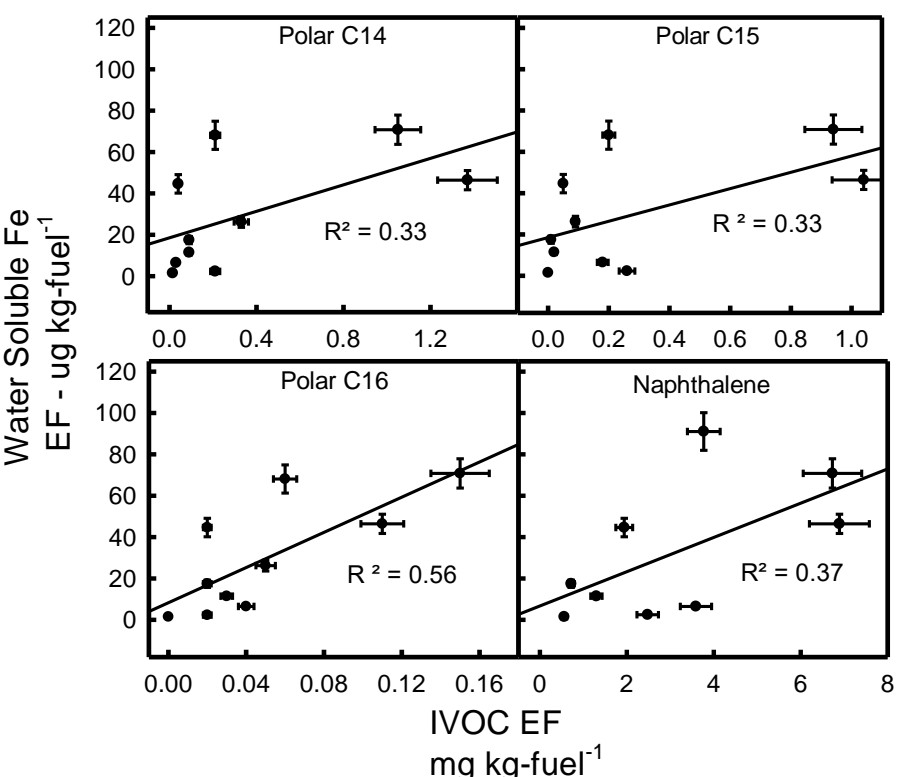






Figure 4: Scatter plots of water-soluble iron versus the sum of IVOCs reported in EF (g kg-fuel⁻
¹).

Finally, the relationship between water-soluble iron and speciated organics, specifically

naphthalene and IVOCs, was examined.  In contrast with all other measured parameters, Figure 4
shows relatively strong correlations between water-soluble iron and some of the IVOC species.
Figure 4 presents the classifications which have the strongest correlation with water-soluble iron.
Water-soluble iron relationships with other IVOCs can be found in the supplementary
information (SI8). The correlation to water-soluble iron is highest for IVOC-polar species with
16 carbons ($R^2 = 0.56$).

As water-soluble iron trends well with naphthalene and polar-IVOCs, but not with bulk

EC or OC, it is highly suggestive that iron solubility from the direct emission samples is
primarily dependent on interactions with the species of carbon present in the particles during the
extraction process. To better understand these interactions, a preliminary laboratory study was
conducted to explore both i) the effect of these organic compounds on iron solubility and ii) the
effect of soluble iron on the oxidation of organic compounds during the extraction process.
Specifically, when naphthalene was added to an insoluble iron source (a soil), iron solubility
increased from 0.8 to 4.2 % of the total, or by a factor of ~5.5, showing that the addition of
naphthalene, alone, can have a significant effect on iron water solubility and that this effect
likely is important during the extraction process.

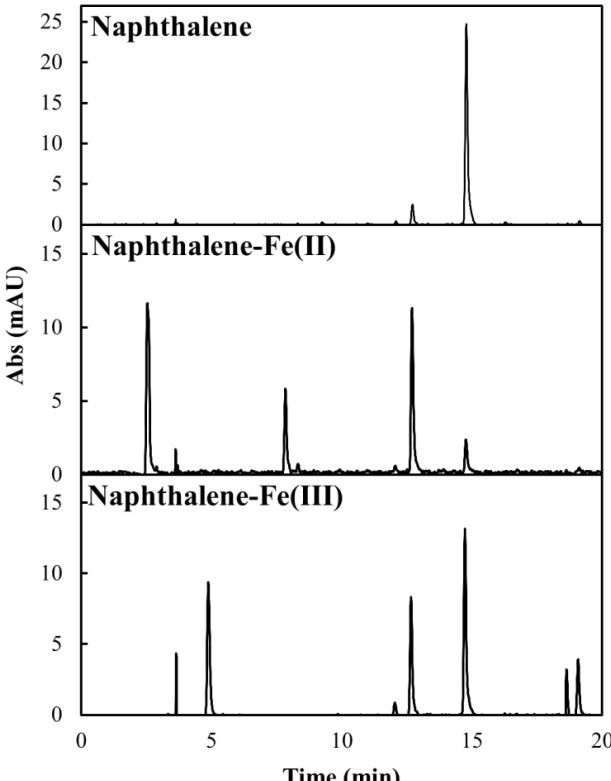


Figure 5: HPLC of resulting reaction between naphthalene and water-soluble iron. Phathalic acid

at 12.5 minutes, phthalic anyhydride at 7.5 minutes, napthol at 15 minutes and naphthalene at 20

minutes. The column uses a C18 stationary phase on beads with 80Å pore size.

Lacking oxidized functional groups, naphthalene was not expected to chelate iron or to,

otherwise, have the ability to increase iron solubility. Thus, we investigated what compounds are

formed from naphthalene during these extractions. Figure 5 shows the new oxidized products

formed from naphthalene during the water extraction. In the presence of soluble iron, HPLC

retention time analysis shows the presence of phthalic acid (12.5 minutes), phthalic anhydride

(7.5 minutes), and naphthol (15 minutes). The peaks at and below 5 min were not identified but,





based on the retention times, these are thought to be low molecular mass, highly polar organic
products and is consistent with other studies (Haynes et al., 2019)
*3.4. Iron-carbon interactions*

There are at least two methods in which organic compounds can lead to increased iron

solubility: a) reduction of Fe(III) to Fe(II) or b) bringing soluble iron into solution via chelation.
The first one is generally achieved by photochemistry (Pehkonen et al., 1993), which is not
directly applicable to this study.  The second, chelation, generally requires oxidized functional
groups as shown in Figure 5. The extent of the ability for phthalic acid (a dicarboxylic acid) to
chelate iron has not been reported, however, it is known that similar molecular mass organic
diacids have significant ability to chelate iron, thus pulling it into solution (Paris and Desboeufs,
2013).  Here, we suggest that the observed correlations between IVOC/naphthalene and water-
soluble iron can be best explained with Fenton reactions, resulting in propagation of radical
reactions (Pehkonen et al., 1993).  As shown from the Fe XANES valance plot, the iron is
predominately Fe(III) (Figure 4). In addition to the Fe(III), it has been shown that $H_2O_2$ forms in
$PM_{2.5}$ water extracts and it been speculated that this formation is from various transition metals
and/or quinones found in $PM_{2.5}$ (Wang et al., 2012).
$Fe^{3+} + H_2O_2 \rightarrow Fe^{2+} + H^+ + HO_2^{\circ}$       (1)
$HO_2^{\circ} \rightarrow H^+ + O_2^{\circ -}$       (2)
$H^+ + O_2^{\circ -} + Naphthalene \rightarrow Oxidized\ Naphthalene$       (3)

In the presence of $H_2O_2$, Fe(III) is known to undergo reaction (1) (Neyens and Baeyens,

2003; Pignatello et al., 2006), resulting in the formation of Fe(II) and $HO_2$ (Pignatello et al.,
2006; Rubio-Clemente et al., 2014), which degrades into superoxide, $O_2^-$, and $H^+$ (2). Superoxide
has the ability to oxidize organic compounds, particularly aromatic structures (3) (Lair et al.,





2008). The resulting structures of these oxidized compounds typically have two oxygen atoms,
which could be arranged in various functional groups (Lair et al., 2008; Rubio-Clemente et al.,
2014), also observed from the HPLC chromatograms. Oxidized single ring aromatic structures
have a strong affinity to iron and have the ability chelate iron into aqueous solution (Hosseini
and Madarshahian, 2009). Based on the laboratory studies of naphthalene and soluble-iron
presented here, naphthalene and/or IVOC oxidation during the extraction process is the most
likely path towards increased iron solubility in primary tailpipe emissions.
**4. Conclusions**

This study shows water-soluble iron is directly formed from vehicle exhaust. The results

show that iron is solubilized in water by specific organic compounds present in automobile
exhaust, and that soluble iron is not necessarily dictated by the overall OC content. Thus, the
implication is that anthropogenic water-soluble iron is a result of chelation from specific organic
compounds, likely their eventual aqueous reaction products. Although the mechanism of these
aqueous transformations were not directly measured in this study, based on Fenton chemistry,
the primary compounds are expected to be oxidized versions of naphthalene and/or IVOCs
(Ledakowicz et al., 1999). Since these oxidation reactions occur fairly quickly (i.e., during the
water extraction), further studies are of interest to better understand how these organic
compounds interact with iron as it enters atmospheric waters and, also, the photo-chemical
interactions between iron and organics.





**Acknowledgements**


The authors thank the excellent and dedicated personnel at the California Air Resources
Board, especially at the Haagen–Smit Laboratory This study was funded by National Science
Foundation grant number 1342599. This research used resources of the Advanced Light Source,
which is a DOE Office of Science User Facility under contract no. DE-AC02-05CH11231.
Financial support was provided by the California Air Resources Board (Contract #12-318). The
California Air Resources Board also provided substantial in-kind support for vehicle
procurement, testing, and emissions characterization.

**Author contribution**


The sample collection scheme was designed by Allen L. Robinson, Allen H. Goldstein
and Brian J. Majestic. Samples were collected by Benton T. Cartledge and Greg T. Drozd.
Organic speciation was performed by Greg T. Drozd. Trace elements were quantified by Joseph
R. Salazar. Iron speciation was performed by Joseph R. Salazar, Rachel York-Marini and Brian
J. Majestic, with the interpretation effort led by Sirine C. Fakra. Bench-top naphthene
experiments were performed by John P. Haynes. Data integration was performed by Joseph R.
Salazar. The manuscript was prepared by Joseph R. Salazar and Brian J. Majestic.







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



**Tables and Figures and Captions**
Table 1: Average of total trace total and water soluble elements from car exhaust reported in EF
($\mu$g kg-fuel$^{-1}$). These samples represent a range of different makes and models of cars. The
values in the parenthesis are the range of the vehicle populualtion. (n=32)
Table 2: Comparison of exhaust composition in g km$^{-1}$ from different dynamometer studies
which included both gasoline and diesel powered light duty vehicles. The values are the mean of
the vehicle population and the values in the parenthesis are the minimum and maximum values.
This table is in g km$^{-1}$ opposed to g kg-fuel$^{-1}$ in Table 1.
















Table 1:

| | Total Elements | Water-Soluble Elements |
|---|---|---|
| **Trace elements (µg kg-fuel$^{-1}$)** | | |
| Na | 50 (0, 200) | 30 (0, 100) |
| Mg | 40 (0, 200) | 8 (0, 60) |
| Al | 100 (0, 2000) | 20 (0, 100) |
| K | 20 (0, 100) | 20 (0, 100) |
| Ca | 200 (0, 1000) | 200 (0, 1000) |
| Ti | 1 (0, 60) | 0.2 (0, 2) |
| V | 0.02 (0, 0.7) | 0.02 (0, 0.7) |
| Cr | 5 (0.04, 20) | 0.6 (0, 4) |
| Mn | 2 (0.02, 10) | 1 (0.007, 8) |
| Fe | 80 (0, 400) | 20 (0, 200) |
| Co | 0.2 (0, 1) | 0.04 (0, 0.7) |
| Ni | 5 (0, 30) | 2 (0, 10) |
| Cu | 20 (0, 200) | 20 (0, 100) |
| Zn | 60 (0, 300) | 40 (0, 300) |
| As | 0.006 (0, 0.03) | 0.006 (0, 0.03) |
| Se | 0.3 (0, 2) | 0.05 (0, 0.5) |
| Rb | 0.2 (0, 0.5) | 0.01 (0, 0.1) |
| Sr | 1 (0.01, 4) | 0.6 (0.003, 3) |
| Mo | 5 (0, 20) | 3 (0.002, 30) |
| Rh | 0.06 (0, 0.5) | 0.007 (0, 0.1) |
| Pd | 0.8 (0, 6) | 0.3 (0, 4) |
| Ag | 0.1 (0, 2) | 0.03 (0, 0.5) |
| Cd | 0.007 (0, 0.3) | 0.009 (0, 0.05) |
| Sb | 0.2 (0, 1) | 0.1 (0, 0.9) |
| Cs | 0.005 (0, 0.02) | 0.002 (0, 0.02) |
| Ba | 5 (0, 20) | 3 (0.06, 20) |
| Ce | 4 (0, 40) | 0.4 (0, 2) |
| Pt | 0.04 (0, 0.4) | 0.01 (0, 0.2) |
| Pb | 0.4 (0, 7) | 0.3 (0, 7) |
| U | 0.002 (0, 0.03) | 0.002 (0, 0.03) |











Table 2:

| | This study Gasoline (n = 32) | Gasoline(Schauer et al., 2002) (n=9) | Gasoline(Norbeck et al., 1998) (n=40) | Diesel(Norbeck et al., 1998) (n=19) |
|---|---|---|---|---|
| Fleet Age | 1990-2014 | 1981-1994 | 1972-1990 | 1977-1993 |
| **PM components (mg km$^{-1}$)** | | | | |
| OC | 1 (0.06, 10) | 3.3 ± 0.21 | 16 ± 32 | 150 ± 330 |
| EC | 10 (0.06, 100) | 0.77 ± 0.023 | 3.5 ± 4.8 | 160 ± 100 |
| sulfate | 0.02 (0.001, 0.1) | 0.08 ± 0.16 | 0.93 ± 1.9 | 0.77 ± .93 |
| **Trace elements (µg km$^{-1}$)** | | | | |
| Ag | 0.01 (0, 0.25) | 4.5 ± 20 | 0 | 0 |
| Al | 10 (0, 110) | 20 ± 17 | 19 ± 37 | 31 ± 75 |
| Ba | 0.6 (0, 4.4) | 0 | 0 | 68 ± 75 |
| Ca | 30 (0, 130) | 26 ± 8.5 | 81 ± 120 | 650 ± 930 |
| Cd | 0.00 (0, 0.04) | 0 | 0 | 0 |
| Co | 0.01 (0,0.25) | - | 0 | 0 |
| Cr | 0.6 (0.008, 4) | 0 | 0 | 6.2 ± 12 |
| Cu | 3 (0, 27) | 0 | 6.2 ± 6.2 | 19 ± 31 |
| Fe | 10 (0, 62) | 8.3 ± 2.3 | 280 ± 680 | 830 ± 1000 |
| K | 2 (0, 15) | 3.0 ± 11.3 | 0 | 50 ± 170 |
| Mg | 7 (0, 120) | - | 25 ± 31 | 99 ± 200 |
| Mn | 0.2 (0.002, 1.3) | 0 | 0 | 6.2 ± 6.2 |
| Mo | 0.5 (0, 3.6) | 2.3 ± 6.8 | 0 | 6.2 ± 12 |
| Ni | 0.6 (0, 5.2) | 0 | 6.2 ± 12 | 12 ± 18 |
| Pb | 0.04 (0, 0.57) | 0 | 25 ± 93 | 19 ± 62 |
| Sb | 0.02 (0, 0.21) | 17 ± 39 | 0 | 0 |
| Sr | 0.1 (0, 0.68) | 0.75 ± 2.3 | 0 | 0 |
| Zn | 7 (0, 37) | 14 ± 1.5 | 110 ± 170 | 810 ± 1500 |



