# Peer review of "Water-soluble iron emitted from vehicle exhaust is linked to primary speciated organic 1 2 compounds 3 Joseph R. Salazar\*, Benton T. Cartledge\*, John P. Haynes\*, Rachel York-Marini\*, Allen L. 4 Robinson‡, Greg T. Drozd€, Allen H. Goldstein¥, Sirine C. Fakra¢"

_Atmospheric Chemistry and Physics, 2019_

## Referee Comment (RC1) · Anonymous Referee #2 · 10 Oct 2019

Review of acp-2019-386

Salazar et al., "Water-soluble iron correlation to primary speciated organics in low-emitting vehicle exhaust"

**Summary:**
The authors present a comprehensive study assessing the iron emitted by a collection of gasoline vehicles spanning a range of emissions certifications. This includes total iron and water-soluble iron as well as complementary analyses to determine the oxidation state of the iron. Interestingly, there is a trend between water-soluble iron emissions and intermediate-volatility organic compound (IVOC) emissions. Through a complementary laboratory study, the authors demonstrate that the iron may interact with some organic compounds, resulting in a transformation to water-soluble iron. Overall, this is a nice paper, and I recommend it for publication in *Atmospheric Chemistry and Physics*, pending adequate response to my comments and those from the other reviewers.

**General Comments:**
Some of the manuscript is unnecessarily repetitive. For example:

> Lines 100-103, lines 115-118, and lines 126-127 are referring to particle sampling and analysis methods. Please combine to a single location within the document.

> Lines 129-131 and lines 142-144 both mention the use of a laminar flow hood for handling of samples. Please remove this redundancy.

In Figures 1, 2, and 4, please use "µg" rather than "ug".

**Specific Comments:**
Lines 61-63: Is the iron present in the gasoline itself, or does it leach from the vehicle components?

Line 118-120: For a field campaign that occurred in 2014, I have a hard time believing that results were published in 2000. Please correct this reference.

Lines 156-159: How was 3% of the filters "measured exactly"? Was this using a filter punch that was precisely 3% of the area of the filters? Please clarify.

Lines 178-182: I may have missed this definition, but what is "µXRF"? Does it differ from a typical X-ray fluorescence measurement?

Lines 235-241: It is a little unclear to me how the total iron emissions are defined. Is this the sum of the water-soluble iron from the water extractions described in Section 2.3 and the remaining iron that underwent the acid digestion in Section 2.4? Or was water-soluble iron determined from one filter and total iron determined from another filter? Please clarify.

Lines 246-248: Why do the authors use the symbol from the periodic table for metals in previous sentence in this paragraph but not here?

Lines 258-261: "Trace elements $km^{-1}$" and "per km emissions" are just distance-based emission factors (as opposed to the fuel-based emission factors that the authors have used). I recommend using "distance-based emission factors" in both of these lines.

Figures 1 and 2: I'm wondering if it could be more informative to present the total iron emissions as, e.g., Figure 1a, and then have Figure 1b include box plots of the water-soluble iron fraction. This is just a thought that could potentially be more informative to drive home how much of the iron is actually water-soluble.

Lines 275-280: I have another thought on the presentation of results here. Given a lack of trend in total iron with emission certification, I'm curious if it would be worth exploring a trend in the ratio of total iron to particulate matter (PM) mass (e.g., $EF_{Fe}/EF_{PM}$). I suspect that the emissions of iron relative to total PM will increase, which could be an interesting result.

Lines 377-387: If I am understanding this correctly, it suggests that Fe(III) is emitted yet is rapidly converted to Fe(II). This may be worth stating explicitly.

---

## Referee Comment (RC2) · Anonymous Referee #1 · 3 Nov 2019

General:

This paper characterizes Fe solubility found in tailpipe emissions from different vehicle classes and proposes a mechanism involving Fenton chemistry with aromatic compounds to explain their results. This work is novel and of interest to the aerosol community. On another note, I found it refreshing to read an ACPD paper that was well-written yet concise. I recommend this paper for publication after addressing my minor comments.

Major Comments:

1. The title is a bit awkward and overly complicated. I suggest that the authors simplify

their title to something like "Water-soluble iron emitted from vehicle exhaust is linked to cyclic organic compounds".

2. The rationale for targeting the IVOCs was not well-explained. I suggest a paragraph in the introduction briefly discussing different organics emitted from vehicles and what their possible role in Fe solubility may be.

Specific Comments:

Abstract:

1. Define EC and OC.

2. Sentence on lines 21-24 needs to be rephrased. I found it confusing.

3. The end of the abstract should more clearly spell out the mechanism for the increased Fe solubility and include the role of Fenton chemistry.

Introduction:

1. I suggest mentioning the different organics found in vehicle exhaust with attention to cyclic compounds and IVOCs. I also suggest mentioning how those compounds could affect Fe solubility to help establish the rationale for that aspect of your work.

2. I encourage the authors to include and discuss the following papers relevant to this study in the introduction and the results section: [Chen and Grassian, 2013; Fu et al., 2012; Meskhidze et al., 2017]

Methods:

1. Define FID

Results:

1. Figure SI1 is important for showing that bulk organics and markers of inorganic acid processing (e.g., sulfate and nitrate) do not correlate with Fe and are not important for Fe solubility. I suggest showing at least these two aspects of your correlation analysis
in the main manuscript and not in the SI.

2. Line 311: Define LFC.

3. Line 324: provide some more rationale for why you targeted these specific organics.

4. Lines 330-336: While these plots are compelling, the authors should provide a sentence or two with some explanation for the scatter in the data.

Conclusions:

1. I suggest that the authors reiterate that Fe solubility was not related to inorganic acid processing. This is a very important point since many studies assume that sulfuric acid, in particular, is the most important acid that induces changes in Fe solubility.

References:

Chen, H. H., and V. H. Grassian (2013), Iron dissolution of dust source materials during simulated acidic processing: The effect of sulfuric, acetic, and oxalic acids, Environmental Science & Technology, 47(18), 10312-10321.

Fu, H. B., J. Lin, G. F. Shang, W. B. Dong, V. H. Grassian, G. R. Carmichael, Y. Li, and J. M. Chen (2012), Solubility of iron from combustion source particles in acidic media linked to iron speciation, Environmental Science & Technology, 46(20), 11119-11127.

Meskhidze, N., D. Hurley, T. M. Royalty, and M. S. Johnson (2017), Potential effect of atmospheric dissolved organic carbon on the iron solubility in seawater, Marine Chemistry, 194, 124-132.

———————————————————

---

## Author Comment (AC1) · 13 Dec 2019

Summary: The authors present a comprehensive study assessing the iron emitted by a collection of gasoline vehicles spanning a range of emissions certifications. This includes total iron and water-soluble iron as well as complementary analyses to determine the oxidation state of the iron. Interestingly, there is a trend between water-soluble iron emissions and intermediate-volatility organic compound (IVOC) emissions. Through a complementary laboratory study, the authors demonstrate that the iron may interact with some organic compounds, resulting in a transformation to water-soluble iron. Overall, this is a nice paper, and I recommend it for publication in Atmospheric

[Figure]

Chemistry and Physics, pending adequate response to my comments and those from the other reviewers.

Reply: We thank the Reviewer for the kind words.

General Comments: Some of the manuscript is unnecessarily repetitive. For example:

Lines 100-103, lines 115-118, and lines 126-127 are referring to particle sampling and analysis methods. Please combine to a single location within the document.

Reply: We thank the reviewer for the suggestion to reduce the repetitiveness, lines 100-103, lines 115-118, and lines 126-127 are combined. Line 110-116 now reads "Emission samples were collected using a constant volume sampler from which a slip-stream of dilute exhaust was drawn at a flow rate of 47 L min-1. Particle phase emissions were collected using three sampling trains operated in parallel off of the end of the CVS dilution tunnel. Train 1 contained a Teflon filter (47 mm, Pall-Gelman, Teflo R2PJ047). Train 2 contained two quartz filters (47 mm, Pall-Gelman, Tissuquartz 2500 QAOUP) in series. Train 3 contained an acid-cleaned Teflon filter followed by a quartz filter (47 mm, Teflo, Pall Life Sciences, Ann Arbor, MI) and the flow rate was 0.5 L min-1 through each Tenax tube." Lines 115-118 and 126-127 were deleted.

Lines 129-131 and lines 142-144 both mention the use of a laminar flow hood for handling of samples. Please remove this redundancy. Reply: To make the manuscript more concise, text in line 142 "and handled inside a polypropylene laminar flow hood (NuAire, Plymouth, MN)" was removed

In Figures 1, 2, and 4, please use "$\mu$g" rather than "ug". Reply: We thank the Reviewer for bringing this to our attention: We changed ug to $\mu$g in Figures 1, 2, and 4

Specific Comments: Lines 61-63: Is the iron present in the gasoline itself, or does it leach from the vehicle components?

Reply: To avoid any misunderstanding, Line 65-69 changed to "Iron is contained in many fuels which has pre-combusted concentrations ranging from 13-1000 $\mu$g L-1 (Lee

and Von Lehmden, 1973; Santos et al., 2011; Teixeira et al., 2007). Within the engine, computational models of combustion in engines suggest that iron emissions could also originate from the fuel injector nozzle inside the engine block (Liati et al., 2015)."

Line 118-120: For a field campaign that occurred in 2014, I have a hard time believing that results were published in 2000. Please correct this reference.

Reply: We thank the Reviewer for catching this. We were using the methods, not the data. Thus, line 131-132 is changed to "procedure for these data presented elsewhere" from "these data are presented elsewhere"

Lines 156-159: How was 3% of the filters "measured exactly"? Was this using a filter punch that was precisely 3% of the area of the filters? Please clarify.

Reply: To clarify how the filters were cut Line 166-167 changed to "~3% of the filters was measured and cut using a ceramic blade" Lines 178-182: I may have missed this definition, but what is "$\mu$XRF"? Does it differ from a typical X-ray fluorescence measurement?

Reply: The $\mu$ refers to the small spot size that the beam was able to fluoresce, thus line 188 has been changed to "micro X-ray fluorescence ($\mu$XRF)" and $\mu$XRF has been added to XRF in line 188 and 191 for consistency

Lines 235-241: It is a little unclear to me how the total iron emissions are defined. Is this the sum of the water-soluble iron from the water extractions described in Section 2.3 and the remaining iron that underwent the acid digestion in Section 2.4? Or was water-soluble iron determined from one filter and total iron determined from another filter? Please clarify.

Reply: Yes, the iron is summed from the water extractions described in Section 2.3 and the remaining iron that underwent the acid digestion in Section 2.4. Line 166-169 has been clarified to "First ~3% (measured exactly) of the filters were cut and saved for X-ray absorption near edge structure (XANES) spectroscopy, then the water-soluble

elements were extracted and, lastly the polymethylpentene ring was removed from the Teflon filters."

Lines 246-248: Why do the authors use the symbol from the periodic table for metals in previous sentence in this paragraph but not here?

Reply: We thank the Reviewer for bringing this to our attention. Elements in line 257 were changed to the names of the elements.

Lines 258-261: "Trace elements km-1" and "per km emissions" are just distance-based emission factors (as opposed to the fuel-based emission factors that the authors have used). I recommend using "distance-based emission factors" in both of these lines.

Reply: We thank the reviewer for the clarification. "Trace elements km-1" and "per km emissions" have been changed to "distance-based emission factors" in Lines 269-272 "Table 2 compares the average exhaust PM composition and trace elements in distance-based emission factors in this study to literature values for other passenger vehicles, including one diesel and three gasoline exhaust studies. For all elements, the distance-based emission factors were greater in the diesel cohort, relative to the gasoline vehicles."

Figures 1 and 2: I'm wondering if it could be more informative to present the total iron emissions as, e.g., Figure 1a, and then have Figure 1b include box plots of the water-soluble iron fraction. This is just a thought that could potentially be more informative to drive home how much of the iron is actually water-soluble.

Reply: This is a great suggestion and we thank the reviewer. Below is the revised graph and the removed graph.

Lines 275-280: I have another thought on the presentation of results here. Given a lack of trend in total iron with emission certification, I'm curious if it would be worth exploring a trend in the ratio of total iron to particulate matter (PM) mass (e.g., EFFe/EFPM). I suspect that the emissions of iron relative to total PM will increase, which could be an

[Figure]

interesting result.

Reply: The authors agree that this could be useful, unfortunately overall PM mass wasn't measured as part of this study.

Lines 377-387: If I am understanding this correctly, it suggests that Fe(III) is emitted yet is rapidly converted to Fe(II). This may be worth stating explicitly.

Reply: Added to Line 406 to restate the above chemistry and clear up any confusion "This overall process suggests that Fe(III) is emitted through car exhaust through interaction with water and organics undergoes a Fenton like reaction and converted to Fe(II) and the iron is chelated by the resulting oxidized organics."
* * *
[Figure]

Figure 2: Water-soluble iron from the 32 vehicles tested reported in water-soluble iron fraction. The center black line represents the median value and the edges of the boxes represent the 25th and 75th percentiles while the whiskers are the 10th and 90th percentiles.

[Figure]

Figure 2: Water-soluble iron from the 32 vehicles tested reported in EF (µg kg-fuel⁻¹). The center black line represents the median value and the edges of the boxes represent the 25ᵗʰ and 75ᵗʰ percentiles while the whiskers are the 10ᵗʰ and 90ᵗʰ percentiles.

**Fig. 1.**

---

## Author Comment (AC2) · 14 Dec 2019

General: This paper characterizes Fe solubility found in tailpipe emissions from different vehicle classes and proposes a mechanism involving Fenton chemistry with aromatic compounds to explain their results. This work is novel and of interest to the aerosol community. On another note, I found it refreshing to read an ACPD paper that was well-written yet concise. I recommend this paper for publication after addressing my minor comments.

Reply: We thank the Reviewer for their comments.

[Figure]

Major Comments: 1. The title is a bit awkward and overly complicated. I suggest that the authors simplify their title to something like "Water-soluble iron emitted from vehicle exhaust is linked to cyclic organic compounds".

Reply: We agree that this would be clearer to the reader, so the title has been changed from "Water-soluble iron correlation to primary speciated organics in low-emitting vehicle exhaust" to "Water-soluble iron emitted from vehicle exhaust is linked to primary speciated organic compounds"

2. The rationale for targeting the IVOCs was not well-explained. I suggest a paragraph in the introduction briefly discussing different organics emitted from vehicles and what their possible role in Fe solubility may be.

Reply: The authors agree that a rational for targeting IVOCs would be beneficial to the introduction. Line 81-89 refers to the research on organic species and solubilized iron. Line 89-92 was added to better explain the rational for targeting IVOCs. Line 81-89 states "A third, broad, iron solubilization hypothesis emphasizes an iron-organic interaction (Baba et al., 2015; Vile et al., 1987). For example, a significant increase in water-soluble iron is observed in the presence of oxalate and formate in ambient aerosols and in cloud droplets (Paris et al., 2011; Zhu et al., 1993). Other studies have suggested that the photolysis of polycyclic aromatic hydrocarbons leads to reduced iron, which may result in greater iron water solubility (Faiola et al., 2011; Haynes et al., 2019; Pehkonen et al., 1993; Zhu et al., 1993)."

Line 89-92: Vehicle exhaust contains many organic species including secondary organic aerosol (SOA) Single-ring aromatic compounds (C6-C9) PAHs, hopanes, steranes, alkanes, organic acids and intermediate volatility organic compound (IVOCs) which are longer chain organic species. (Cheung et al., 2010; Zhao et al., 2016)

Specific Comments: Abstract:

1. Define EC and OC. Reply: For clarification, Line 22-23 now reads "elemental carbon

(EC), organic carbon (OC)"

2. Sentence on lines 21-24 needs to be rephrased. I found it confusing. Reply: We thank the reviewer for this suggestion. To clarify for the reader, Line 23-26 has been divided in to two sentences and changed to "Naphthalene and intermediate volatility organic compounds (IVOC) were quantified for a subset of vehicles. The IVOC quantified contained 12 to 18 carbons and were divided into three subgroups: aliphatic, single ring aromatic (SRA), and polar (material not classified as either aliphatic or SRA)."

3. The end of the abstract should more clearly spell out the mechanism for the increased Fe solubility and include the role of Fenton chemistry.

Reply: To clarify for the reader, line 34-37 has been changed to "These results suggest that the large driver in water-soluble iron from primary vehicle tail-pipe emissions is related to the organic composition of the PM. We hypothesize that, during the extraction process, specific components of the organic fraction of the PM are oxidized and chelate the iron into water"

Introduction:

4. I suggest mentioning the different organics found in vehicle exhaust with attention to cyclic compounds and IVOCs. I also suggest mentioning how those compounds could affect Fe solubility to help establish the rationale for that aspect of your work.

Reply: 81-89 refers to the research on organic species and solubilized iron which led to the rationale to establish a relationship between water soluble iron and IVOCs. Line 81-89 states "A third, broad, iron solubilization hypothesis emphasizes an iron-organic interaction (Baba et al., 2015; Vile et al., 1987). For example, a significant increase in water-soluble iron is observed in the presence of oxalate and formate in ambient aerosols and in cloud droplets (Paris et al., 2011; Zhu et al., 1993). Even when compared to sulfuric acid, oxalic acid results in a greater increase in iron solubility because of the organic iron interaction (Chen and Grassian, 2013). Other studies have suggested that the photolysis of polycyclic aromatic hydrocarbons leads to reduced iron, which may result in greater iron water solubility (Faiola et al., 2011; Haynes and Majestic, 2019; Haynes et al., 2019; Pehkonen et al., 1993; Zhu et al., 1993)." Line 89-92: Vehicle exhaust contains many organic species including secondary organic aerosol (SOA) Single-ring aromatic compounds (C6-C9) PAHs, hopanes, steranes, alkanes, organic acids and intermediate volatility organic compound (IVOCs) which are longer chain organic species. (Cheung et al., 2010; Zhao et al., 2016)

5. I encourage the authors to include and discuss the following papers relevant to this study in the introduction and the results section: [Chen and Grassian, 2013; Fu et al., 2012; Meskhidze et al., 2017]

Reply: We agree that these references are important for this topic, so the following lines have been added. Line 57 "From these combustion sources, it has been shown that the species of iron differed greatly and had an impact in iron solubility (Fu et al., 2012)." Line 84 "Even when compared to sulfuric acid, oxalic acid results in a greater increase in iron solubility because of the organic iron interaction (Chen and Grassian, 2013)" Meskhidze et al., 2017, while important, was not added to the manuscript because a discussion of iron dissolved in seawater might confuse readers in understanding the context of this manuscript.

Methods:

1. Define FID

Reply: For clarity, Line 128-129 now reads "by gas chromatography, with detection by a flame ionization detector (FID)."

Results:

1. Figure SI1 is important for showing that bulk organics and markers of inorganic acid processing (e.g., sulfate and nitrate) do not correlate with Fe and are not important for Fe solubility. I suggest showing at least these two aspects of your correlation analysis

in the main manuscript and not in the SI.

Reply: We agree with this suggestion and a plot showing bulk organic carbon (OC) and sulfate is now Figure 3 in the manuscript and referenced in line 302 and 307.

2. Line 311: Define LFC.

Reply: LCF is defined in section 2.6 Line 191 in the methods "Least-square linear combination fitting (LCF) was subsequently performed in the range 7090 to 7365 eV to confirm iron valence and further identify the major mineral groups present."

3. Line 324: provide some more rationale for why you targeted these specific organics.

Reply: To clarify for the reader, Line 340 was modified to "leading to the investigation of organic species which resulted in a correlation to longer chain IVOC and naphthalene (Haynes and Majestic, 2019)"

4. Lines 353-354: While these plots are compelling, the authors should provide a sentence or two with some explanation for the scatter in the data. Reply: To address the scatter in the manuscript line 336 was added "The variance of figure 4 could result from the fact that, in addition to the IVOCs, other factors also influence iron water solubility"

Conclusions: 1. I suggest that the authors reiterate that Fe solubility was not related to inorganic acid processing. This is a very important point since many studies assume that sulfuric acid, in particular, is the most important acid that induces changes in Fe solubility.

Reply: We agree that this is one of the key points of the paper. Thus, we have changed line 410 to "This study shows water-soluble iron is directly formed from vehicle exhaust and not correlated to sulfates"

[Figure]

Figure 3: Linear correlation plots representing EF in mg kg-fuel-1 for sulfate and organic carbon (OC) in µg kg-fuel-1 for water-soluble iron. Correlation lines and R2 values for all elements are shown.

**Fig. 1.**